# Characterization of Four New Compounds from *Protea cynaroides* Leaves and Their Tyrosinase Inhibitory Potential

**DOI:** 10.3390/plants11131751

**Published:** 2022-06-30

**Authors:** Masande Yalo, Masixole Makhaba, Ahmed A. Hussein, Rajan Sharma, Mkhuseli Koki, Ndikho Nako, Wilfred T. Mabusela

**Affiliations:** 1Department of Chemistry, University of the Western Cape, Private Bag X17, Bellville 7535, South Africa; masandeyalo@gmail.com (M.Y.); makaba.v@gmail.com (M.M.); mkhuselikoki60@gmail.com (M.K.); ndikhonako@gmail.com (N.N.); 2Chemistry Department, Cape Peninsula University of Technology, Symphony Rd., Bellville 7535, South Africa; mohammedam@cput.ac.za (A.A.H.); sharmar@cput.ac.za (R.S.)

**Keywords:** Proteaceae, *Protea cynaroides*, phytochemistry, NMR spectroscopy, tyrosinase

## Abstract

*Protea cynaroides* (king protea) is a flowering plant that belongs to the Proteaceae family. This multi-stemmed shrub is the national flower of South Africa and has important economic and medicinal values. Traditionally, the main therapeutic benefits of this plant species include the treatment of cancer, bladder, and kidney ailments. There are very limited reports on the isolation of phytochemicals and their biological evaluation from *P. cynaroides*. In this study, the leaves of *P. cynaroides* were air-dried at room temperature, powdered, and extracted with 80% methanol followed by solvent fractionation (hexane, dichloromethane, ethyl acetate, and butanol). The ethyl acetate and butanol extracts were chromatographed and afforded four new (**1**–**4**) and four known (**5**–**8**) compounds, whose structures were characterized accordingly as 3,4-bis(4-hydroxybenzoyl)-1,5-anhydro-D-glucitol (**1**), 4-hydroxybenzoyl-1,5-anhydro-D-glucitol (**2**), 2-(hydroxymethyl)-4-oxo-4*H*-pyran-3-yl-6-*O*-benzoate-β-D-glucopyranoside (**3**), 3-hydroxy-7,8-dihydro-β-ionone 3-*O*-β-D-glucopyranoside (**4**), 4-hydroxybenzoic acid (**5**), 1,5-anhydro-D-glucitol (**6**), 3,4-dihydroxybenzoic acid (**7**), and 3-hydroxykojic acid (**8**). The structural elucidation of the isolated compounds was determined based on 1D and 2D NMR, FTIR, and HRMS spectroscopy, as well as compared with the available literature data. The tyrosinase inhibitory activities of the extracts and isolated compounds were also determined. According to the results, compounds **7** and **8** exhibited potent competitive tyrosinase inhibitory activity against L-tyrosine substrates with IC_50_ values of 0.8776 ± 0.012 and 0.7215 ± 0.090 µg/mL compared to the control (kojic acid, IC_50_ = 0.8347 ± 0.093), respectively. This study is the first chemical investigation of compounds **1**–**4** from a natural source and the first report of the biological evaluation of compounds **1**–**5** against the tyrosinase enzyme. The potent anti-tyrosinase activity exhibited by *P. cynaroides* constituents will support future exploration of the plant in the cosmetic field upon further biological and clinical investigations.

## 1. Introduction

Tyrosinase is a multifunctional copper-containing enzyme; it plays diverse physiological roles in different organisms [1]. Tyrosinase catalyzes the initial step in the formation of the pigment melanin from tyrosine, it is thought to be involved in wound healing and possibly sclerotization of the cuticle [2] in plants. In addition, tyrosinase is known to be involved in the molting process of insects and the adhesion of marine organisms [3,4,5,6]. The various phenolic compounds involve the physiological substrates of tyrosinase and oxidize in the observed browning pathway when the tissues are damaged [7]. In mammals, tyrosinase overactivity causes the accumulation of melanin pigments [8]. Excessive amounts of melanin darken the skin and cause the formation of dark spots [9]. The skin-lightening agents of many cosmeceutical products contain tyrosinase inhibitors. A variety of skin whitening agents have been described, such as ascorbic acid, kojic acid, hydroquinone, glutathione, arbutin, and cysteine [10]. Many cosmetic products for the treatment of hyperpigmentation in the market are formulated and contain one or more of the above-mentioned compounds. The majority of the tyrosinase inhibitors demonstrated serious side effects, such as kojic acid and hydroquinone [10,11]. New natural, safe, and efficient skin lightening agents are demanded by people of different ethnic groups and are highly appreciated by the cosmetic industry [12,13,14].

The Proteaceae family contains more than 1700 species belonging to 83 genera [15]. They often grow in places where there are dry climates, Mediterranean-type, and in Southern Hemisphere climates, with a very large number of species in South Africa and Australia [16,17,18]. Proteaceae are grown at a large scale in South Africa, Zimbabwe, Australia, Israel, and New Zealand, for export, particularly to the United States, an important economic market for their cut flowers [19]. There are few scientific reports on the chemistry and biological evaluations of Proteaceae. Recently, the crude extract of *Dilobeia thouarsii*, a member of the Proteaceae family, was reported to have antimicrobial effects in vitro. The high antimicrobial activity of *D. thouarsii* leaf extracts against *S. aureus* supports its traditional use to treat skin infections in Madagascar [20].

*Protea cynaroides* (king protea) is an upright and multi-stemmed shrub that grows between 0.3 and 2 m tall. It has thin branches, with hairless stems [21]. The leaves are curved, oval, or narrowly elliptic, ranging from 50 to 120 mm in length and 50 to 75 mm in width. The flowerhead sizes range from 120 mm to 300 mm in diameter and the color of the bracts, which are either hairy or hairless, range from pink to creamy-white [22]. *P*. *cynaroides* has one of the widest distribution ranges of all the Proteaceae and occurs from the Cedarberg in the northwest to Grahamstown in the east. It occurs on all mountain ranges in this area, except for the dry interior ranges, and at all elevations from sea level to 1500 m high. According to the literature, *P*. *cynaroides* is used primarily to treat cancer, bladder, and kidney ailments [23]. Verotta et al. [17] also reported that the plant tissue has been used for a range of applications, including in leather tanning, the production of wagon wheels (*P. nitida*), and cough syrup. There are very limited reports available on the chemical substances present in the *Protea* genus. However, a few phenolic compounds [17,24,25], polyol sugars [26], and aromatic esters (aryl glycosides) have been identified from the leaves of some species [27]. Wu et al. [28] reported the isolation of 3,4-dihydroxybenzoic acid from *P. cynaroides* along with other phenolic compounds, such as caffeic, ferulic, gallic, and salicylic acids. In a recent study conducted on the investigation of the polyphenol composition of *Protea* pure (*P. cynaroides* and *P. neriifolia*) and hybrid cultivars (Black beauty and Limelight), 41 phenolic acid esters and 25 flavonoid derivatives, including 5 anthocyanins, an undescribed hydroxycinnamic acid-polygalatol ester, and caffeoyl-*O*-polygalatol (1,5-anhydro-[6-*O*-caffeoyl]-sorbitol(glucitol)) were identified [29]. Several plants in the Proteaceae family have been reported to possess melanin inhibition potentials. For instance, the polar extracts of the root of *P. madiensis* Oliv. were shown to inhibit melanogenesis when used traditionally as herbal medicine for the treatment of skin diseases [30].

Considering the economic and medicinal importance of *P. cynaroides* in South Africa, especially in Cape Town, a phytochemical investigation of the methanol leaf extract of this plant was explored. In addition, the tyrosinase inhibitory activity of the fractions (hexane, dichloromethane, ethyl acetate, and butanol) and isolated compounds were also determined.

## 2. Results and Discussion

### 2.1. Chemical Characterization

Repeated silica gel and Sephadex column chromatography of EtOAc and BuOH extracts of *P. cynaroides* yielded four new and four known compounds (Figure 1). The structures of compounds **1**–**4** were elucidated based on 1D (^1^H, ^13^C, and DEPT 135) and 2D NMR experiments (HSQC and HMBC), FTIR, and LC–MS data, while the known compounds **5**–**8** were confirmed by comparing the spectroscopic data to the literature data.

#### Structural Elucidation of the Isolated Compounds

Compound **1** was obtained as light-brown crystals. The molecular formula was established as C_20_H_20_O_9_ by the positive-ion HRESI-MS, which showed molecular ion [M+H]^+^ at *m*/*z* 405.1195 (calc. for 404.3742). The IR spectrum of **1** showed absorption bands for the hydroxyl groups (3335 cm^−1^), conjugated ester (1693 cm^−1^), α, β-unsaturated carbonyl (1633 cm^−1^), and aromatic C=C (1600 cm^−1^) functionalities. The ^1^H NMR data (Table 1, Appendix A) of compound **1** indicated the presence of eight protons resonating as doublets in the aromatic region at ẟ_H_ 7.71 (2H, *d*, *J* = 8.8 Hz, H-2/H-6), 7.68 (2H, *d*, *J* = 8.8 Hz, H-2″/H-6″), and 6.78 (4H, *d*, *J* = 8.8 Hz, H-3/3′, H-5/5′), indicating the presence of two disubstituted benzene moieties. The other protons resonated ẟ_H_ 5.27 (1H, *t*, *J* = 9.3, H-3), 5.05 (1H, *t*, *J* = 9.7, H-4), 3.94 (1H, *dd*, *J* = 11.1, 5.5, H-1b), 3.76 (1H, *m*, H-2), 3.59 (1H, *ddd*, *J* = 4.2, 2.3, 2.2, H-5), 3.45 (1H, H-6b), 3.38 (1H, H-6a), and 3.33 (1H, *dd*, *J* = 11.1, 5.5, H-1a), showing the presence of a sugar moiety. The absence of the anomeric proton suggested that the sugar moiety was a 1,5-anhydro-D-glucitol [31]. The ^13^C NMR spectrum of compound **1** showed sixteen carbon signals. Apart from the 4-hydroxybenzoate carbon signals, six oxygenated carbon signals at ẟ_C_ 79.5 (C-5), 77.5 (C-3), 69.9 (C-1), 69.7 (C-4), 68.3 (C-2), and 61.2 (C-6) were observed in the ^13^C NMR spectrum, which support the presence of a sugar moiety (Table 1). The other carbon signals were observed at ẟ_C_ 115.6 (C-3′/C-5′), 115.7 (C-3′’/C-5″), 120.1 (C-1″, 120.8 (C-1′), 131.9 (C-2′’/6″), 132.0 (C-2′/C-6′), 162.3 (C-4′), 162.6 (C-4″), 165.0 (C-7″), and 165.6 (C-7′), which were all in duplicates, indicating that there were two 4-hydroxybenzoate moieties.

The combined spectroscopic analysis of the COSY, HSQC, and HMBC data allowed the establishment of the structure of compound **1** (Figure 2). The HSQC spectra allowed the assignment of all the protons attached to their corresponding carbons. Inspection of ^1^H-^1^H COSY was observed between the two aromatic doublets at H-2′/2″, H-6′/6″, and H-3′/3″, H-5′/5″, which also confirmed the presence of a *para*-substituted benzene ring. In addition, the attachment of the acyl groups to the sugar moiety was confirmed by the HMBC cross-peak correlation between H-3/C-7′ and H-4/H-7″; thus, the structure of compound **1** was established as *3*,4-*bis*(4-hydroxybenzoyl)-1,5-anhydro-D-glucitol. A Sci-Finder database search provided no evidence for compound **1** as having been previously reported; therefore, it was proposed as a new compound. Similar structures have been isolated before but with a pyrogallol moiety instead of a 4-hydroxybenzoate. These kinds of compounds are known to show different bioactivities such as alpha-glucosidase, alpha-amylase inhibition, and antioxidant activities [32].

Compound **2** was obtained as white crystals. The molecular formula was established as C_13_H_16_O_7_ by the positive-ion HRESI-MS, which showed molecular ion [M+H]^+^ at *m*/*z* 285.0978 (calc. for 284.3659). Compound **2** was somewhat similar to compound **1,** indicating that the structures of both compounds were closely related, and the only difference was likely the absence of the 4-benzoate moiety in compound **2**. The ^1^H NMR data indicated the presence of two intense protons resonating as doublets in the aromatic region at ẟ_H_ 7.83 (2H, *d*, *J* = 8.8 Hz, H-2′/H-6′), and 6.75 (2H, *d*, *J* = 8.8 Hz, H-3′/H-5′) indicating the presence of 4-hydroxybenzoate moiety. The other protons resonated ẟ_H_ 4.81 (1H, *t*, *J* = 9.3 Hz, H-4), 3.89 (1H, *dd*, *J* = 11.2, 4.6, H-1b), 3.51 (1H, *m*, H-2), 3.51 (1H, *m*, H-3), 3.49 (1H, *m*, H-6b), 3.40 (1H, *m*, H-6a), 3.38 (1H, *m*, H-5), and 3.16 (1H, *t*, *J* = 10.6 Hz, H-1a), reminiscent of the presence of a sugar or open oxygenated chain moiety. Inspection of the ^13^C NMR data showed signals indicative of the 4-hydroxybenzoyl moiety resonating at δ_C_ 167.7 (C-7′), 163.8 (C-4′), 133.2 (C-3′/5′), 122.1 (C-1′), and 116.3 (C-2′/6′). Apart from the 4-hydroxybenzoyl carbon signals, six oxygenated carbon signals at δ_C_ 81.0 (C-5), 77.9 (C-3), 73.0 (C-4), 71.7 (C-2), 71.1 (C-1), and 62.9 (C-6) were observed in the ^13^C NMR spectrum, which also supported the presence of a sugar moiety. A further combined analysis of the COSY, HSQC, and HMBC spectra (Appendix A) allowed the establishment of the structure of compound **2**. The HSQC spectrum allowed the assignment of all the protons attached to their corresponding carbons. The ^1^H-^1^H COSY spectrum (Figure 2) revealed a connection between the doublet appearing at ẟ_H_ 7.83 ppm (H-2′/6′) and the doublet at ẟ_H_ 6.75 ppm (H3′/5′), which confirmed the presence of the disubstituted benzene moiety. Other ^1^H-^1^H correlations between ẟ_H_ 3.16 (assigned as H-1a), ẟ_H_ 3.89 (H-1b), and H-2, ẟ_H_ 3.38 (H-5) with 4.81 ppm, already assigned as H-4, confirmed the presence of the sugar moiety. The HMBC correlations between H-4 and the ester carbonyl (C-7′) indicated that the 4-hydroxybenzoate group was linked at C-4 of the 1,5-anhydro-glucitol moiety; thus, the structure of compound **2** was established as 4-hydroxybenzoyl-1,5-anhydro-D-glucitol. A SciFinder database search provided no evidence for compound **2** as having been previously reported; therefore, it was proposed as a new compound.

Compound **3** was obtained as a light brown solid. The molecular formula was established as C_19_H_20_O_11_ by the positive-ion HRESI-MS, which showed the molecular ion [M+H]^+^ at *m/z* 425.1094 (calc. for 425.3689). The IR spectrum of **3** showed absorption bands for hydroxyl groups (3225 cm^−1^), conjugated ester (1688 cm^−1^), unsaturated carbonyl (1644 cm^−1^), and aromatic C=C (1612 cm^−1^) functionalities.

The ^1^H NMR spectrum (Table 2, Appendix A) showed the presence of one set of *o*-coupled aromatic protons at δ_H_ 7.97 (1H, *d*, *J* = 5.6 Hz, H-6) and δ_H_ 6.31 (1H, *d*, *J* = 5.6 Hz, H-5), including oxymethylene protons at δ_H_ 4.69 (1H, *d*, *J* = 14.1 Hz, H-7a) and 4.47 (1H, *d*, *J* = 14.1 Hz, H-7b) that were characteristics of a 1,2-disubstituted 4-pyrone unit. Another set of *o*-coupled aromatic protons appeared at δ_H_ 7.84 (2H, *d*, *J* = 8.8 Hz, H-2″/H-6″) and δ_H_ 6.83 (2H, *d*, *J* = 8.8 Hz, H-3″/H-5″) and were indicative of a 1,4-disubstituted benzoate unit. The sugar moiety was deduced by the presence of an anomeric proton at δ_H_ 4.94 (1H, *d*, *J* = 7.7 Hz) and other characteristic signals, which appeared in the sugar region around δ_H_ 3.12–3.55. The ^13^C and DEPT-135 NMR spectra showed that compound **3** had 17 signals counted for 18 carbons and were assigned as follows: the resonances at δ_C_ 167.9 (C-7″), 163.7 (C-4″), 133.0 (C-2″), 122.1 (C-1″), and 116.4 (C-3″) were characteristic of a *p*-hydroxybenzoyl moiety. The signals of the pyranone moiety appeared at δ_C_ 177.4 (C-4), 163.9 (C-2), 157.5 (C-6), 142.3 (C-3), and 117.7 (C-5). The resonances of the glucopyranosyl unit appeared at δ_C_ 104.0 (C-1′), 77.8 (C-3′), 76.2 (C-5′), 75.4 (C-2′), 71.7 (C-4′), and 64.4 (C-6′). The oxymethylene signal appeared at δ_C_ 57.7 (C-7). The combined spectroscopic analysis of the ^1^H–^1^H COSY, ^1^H–^13^C HSQC, and ^1^H–^13^C HMBC data allowed the establishment of the structure of compound **3**. In the COSY spectra, correlations between δ_H_ 6.31 (H-5) and δ_H_ 7.97 (H-6) confirmed the presence of a 1,2-disubstituted 4-pyrone unit. In addition, a strong correlation was observed between δ_H_ 7.84 (H-2″/6″) and δ_H_. 6.83 (H-3″/5″), which also confirmed the presence of a *para*-substituted benzene ring. The observed correlation of H-6′ from the sugar unit with C-7″ in the HMBC spectrum indicated that the 4-hydroxybenzoyl moiety was attached at C-6′ of the glucosyl. Furthermore, the HMBC experiment revealed a strong correlation of the anomeric proton H-1′ with the carbon signal at δ_C_ 142.3, which was attributed to C-3 of the pyranone unit. In addition, the oxymethylene was located at C-2 of the pyranone unit and was confirmed through HMBC correlations between H-7 and C-2/C-3 (Figure 3). Therefore, compound **3** was determined as 2-(hydroxymethyl)-4-oxo-4*H*-pyran-3-yl-6-*O*-benzoate-β-D-glcopyranoside. A SciFinder database search provided no evidence for compound **3** as having been previously reported; therefore, it is hereby proposed as a new compound.

Compound **4** was obtained as white crystals. The molecular formula was established as C_19_H_32_O_8_ by the positive-ion HRESI-MS, which showed a molecular ion [M+H]^+^ at *m*/*z* 389.2177 (calc. for 388.4615). The IR spectrum of **4** showed absorption bands characteristic of a hydroxyl group (3179 cm^−1^), conjugated ester (1693 cm^−1^), α, β-unsaturated carbonyl (1633 cm^−1^), and aromatic C=C (1600 cm^−1^) functionalities.

The ^1^H NMR spectrum (Table 2) showed signals of two tertiary methyl protons at δ_H_ 0.86 (3H, *s*, H-12), 1.01 (3H, *s*, H-11), a vinyl methyl at 1.56 (3H, *s*, H-13), and an acetyl methyl at δ_H_ 2.07 (3H, *s*, H-10). Three methylene protons were observed resonating at δ_H_ 2.00 (1H, *dd*, *J* = 17.4, 6.4 Hz, H-4a), 2.31 (1H, *dd*, *J* = 17.4, 6.4 Hz, H-4b), 2.09 (1H, *m*, H-7a), 2.16 (1H, *m*, H-7b), and 2.45 (2H, *t*, *J* = 8.1 Hz, H-8). In addition, two oxymethine protons appeared at δ_H_ 3.16 (1H, *m*) and 3.70 (1H, *m*) and were attributed to H-2 and H-3, respectively. Another resonance appeared at δ_H_ 4.28 (1H, *d*, *J* = 7.8 Hz) for the anomeric proton; the *J* coupling constant was consistent with a beta anomeric proton. The remaining sugar resonances appeared as a cluster between δ_H_ 3.0 and 3.70. The ^13^C and DEPT-135 NMR spectra Appendix A showed that compound **4** had 19 carbon resonances, including four methyl groups at δ_C_ 19.5 (C-13), 22.0 (C-12), 25.6 (C-11), and 30.1 (C-10), three methylene at δ_C_ 22.2 (C-7), 37.9 (C-4), and 43.9 (C-8), two oxymethine at δ_C_ 75.7 (C-3) and 76.9 (C-2), and four fully substituted carbon resonances at δ_C_ 42.2 (C-1), 123.9 (C-5), 136.0 (C-6), including one carbonyl at δ_C_ 208.9 (C-9) for the aglycone group. The ^13^C NMR displayed a characteristic anomeric carbon resonance at δ_C_ 101.4 (C-1′), which corresponded to a typical anomeric proton resonance at δ_H_ 4.28 (*d*, *J* = 7.8 Hz) in the HSQC spectrum. This suggested that compound **4** had one sugar group. The remaining sugar moiety signals appeared at δ_C_ 61.4 (C-6′), 70.4 (C-4′), 73.7 (C-2′), 77.2 (C-3′), and 76.8 (C-5′). The coupling of H-3 with H-2 and H-4, as well as H-7 with H-8, were observed in the ^1^H-^1^H COSY spectrum. In the HMBC spectra, there was a strong correlation of the anomeric proton H-1′ with the carbon signal at δ_C_ 75.7 attributed to C-3 of the aglycone unit, which confirmed the attachment position of the sugar moiety. Furthermore, the HMBC experiment exhibited a strong correlation of the oxymethine proton placed at H-2 with C-11, C-12, and C-3. In addition, a correlation between H-13 with C-4, C-5, and C-6 confirmed the position of the methylene at position 4 of the structure. Therefore, based on the above information, compound **4** was determined as 2-hydroxy-7,8-dihydro-ionone-3-*O*-β-D-glucopyranoside. Similarly, a SciFinder database search provided no evidence for compound **4** as having been previously reported; therefore, it is proposed as a new compound. This compound is an isomer of a known compound called 3,4-dihydroxy-7,8-dihydro-ionone-3-*O*-β-D-glucopyranoside (icariside B_8_), which was first isolated from *Epimedium diphyllum* [33].

The known compounds isolated in this study (from *P*. *cynaroides*) were identified by comparing their ^1^H and ^13^C-NMR data with the available literature as 1,5-anhydro-D-glucitol (**5**) [31], 4-hydroxybenzoic acid (**6**) [34], 3,4-dihydrobenzoic acid (**7**) [35], and 3-hydroxykojic acid (**8**) [36,37]. The previously reported compounds from this plant were not isolated but detected using automated chromatographic techniques, such as HPLC.

### 2.2. Inhibitory Activities of Isolated Compounds on Tyrosinase

The average tyrosinase inhibition percentages of the isolated compounds screened at 0.2 and 0.1 mg/mL revealed that compounds **7** and **8** had inhibitions of 100%, while **5** and **6** showed moderate (25%) and weak (55%) inhibitions, respectively. The rest of the compounds were not effective inhibitors of mushroom tyrosinase with their inhibition percentages less than 10%. The compounds with the highest inhibition percentages were investigated further to determine their IC_50_ values. Compounds **7** and **8** (Table 3, Figure 4) showed IC_50_ values of 0.8776 ± 0.12 and 0.7215 ± 0.09 µg/mL, respectively, and the two compounds showed similar inhibition activities compared to kojic acid, a well-known tyrosinase inhibitor. Indeed, benzoic acid and its derivatives were reported as good tyrosinase inhibitors [38], as benzoic acid can chelate the copper at the active site of the enzyme.

Kojic acid is a well-known tyrosinase inhibitor by forming a chelate with the copper ion in the tyrosinase active site through the 5-hydroxyl and 4-carbonyl groups; interestingly, compound **8** has an extra hydroxyl group in position 3, which may have a role in increasing the chelation power. Interestingly, the additional 3-OH group in compound **8** does not appear to have major effects on the inhibitory activity compared to kojic acid. However, further studies on the stability, safety, and toxicity of compounds **7** and **8** will be required for comparison with kojic acid as a potential inhibitor of tyrosinase. Worthy to mention, compound **3**, the glycosylated derivative of kojic acid displayed a low inhibitory activity against tyrosinase, which may reflect the importance of having a free 3-OH group (or 5-OH) to provide extra stability for the inhibition of the tyrosinase enzyme.

## 3. Materials and Methods

### 3.1. Plant Material

The leaves of *Protea cynaroides* were collected at Kirstenbosch National Botanical Gardens, South Africa, Cape Town (−33°59′13.19″ S, 18°25′29.39″ E) on 31 August 2018. The identity of the plant species (voucher number: MY-2018-3) was confirmed by the curator of the Compton Herbarium, Kirstenbosch.

### 3.2. Equipment and Chemical Reagents

The 1D (^1^H, ^13^C, and DEPT-135) and 2D NMR (COSY, HSQC, HMBC) spectra were recorded on the Avance 400 MHz NMR spectrometer (Bruker, Rheinstetten, Germany) at 400 (proton, ^1^H) and 100 (carbon, ^13^C) MHz. Chemical shifts were reported in parts per million (ppm) and coupling constants (*J*) in Hz. The ^1^H and ^13^C NMR values were relative to the internal standard TMS and were acquired in CD_3_OD, CDCl_3_, or DMSO-*d*_6_. HRESI-MS were obtained on a Waters Synapt G2 mass spectrometer (Cone Voltage 15 V), which operated in the negative and/or positive ion modes using direct injection. ATR-FTIR (PerkinElmer Spectrum 100, Llantrisant, Wales, UK) at a transmission mode of 400–4000 cm^−1^ column chromatography was performed using Sephadex (LH-20, Sigma-Aldrich, St. Louis, MO, USA), and normal-phase silica gel 60 (70–230 mesh ASTM, Merck, Readington Township, NJ, USA). TLC was performed on silica gel aluminum sheets (Silica gel 60 F254, Merck) to monitor the fractions. Visualization was achieved with 10% H_2_SO_4_ and detection with the vanillin sulfuric acid reagent and heating to 105 °C.

### 3.3. Extraction and Fractionation of the Plant Material

The air-dried leaves of the fresh plant material (445 g) were blended and extracted by maceration with 80% methanol (1.5 L × 24 h × 3) at room temperature (±25 °C). The methanol extract was filtered and evaporated to dryness under reduced pressure at 40 °C. The total extract was concentrated under a vacuum to remove the methanol for freeze-drying. The freeze-dried material was suspended in water and partitioned successively with n-hexane, DCM, EtOAc, and BuOH. Each extract was concentrated to dryness under reduced pressure. Purification and isolation of natural products were achieved through one or a combination of chromatographic techniques.

The EtOAc extract (3.27 g) was pre-adsorbed on silica gel and fractionated on a column by gravity elution using the mixture of DCM elution as follows: 1 L of 100%, then 1 L volumes of mixtures with EtOAc in the following ratios (80:20), (60:40), (40:60), (20:80), (10:90), and finally 100% EtOAc. Finally, MeOH was introduced in the mixture with EtOAc up to 15% MeOH, (95:5), (90:10), (85:15). A total of thirty-one fractions were analyzed by TLC using CHCl_3_: MeOH: H_2_O (200:52:6). Fractions with similar retention factor (Rf) values were combined. The combined fractions **F_9_**–**F_12_** (55 mg) and **F_19_**–**F_21_** (1.5 g) were pre-adsorbed on silica gel and loaded onto a column for further separation. Fractions **F_9_**–**F_12_** were subjected to a silica gel column and eluted with DCM: MeOH (90:10) isocratically to yield a total of nineteen subfractions, which afforded compound **6** as white crystals (15.0 mg). Fractions **F_19_**–**F_21_** were also subjected to a silica gel column and eluted with DCM: MeOH (90:10) isocratically to afford twenty-four subfractions, which resulted in the isolation of compound **1** as light-brown crystals (0.98 mg).

The BuOH extract (6.82 g) was pre-adsorbed on silica gel and fractionated on a column by gravity elution using the mixture of DCM elution as follows: 1L of 100%, then 1L volumes of mixtures with EtOAc in the following ratios (50:50), (40:60), (30:70), (20:80), (10:90), and 100% EtOAc. Finally, MeOH was introduced in the mixture with EtOAc up to 15% MeOH, (95:5), (90:10), and (85:15). A total of thirty-five fractions were analyzed by TLC using DCM: MeOH: H_2_O (90:10). Fractions with similar retention factor (Rf) values were combined. The combined fractions **F_6_**–**F_9_** (95.0 mg), **F_12_-F_14_** (210.0 mg), **F_16_** (305.0 mg), **F_18_**–**F_20_** (5.10 mg), **F_22_** (150.0 mg), and **F_24_** (75.0 mg) were each pre-adsorbed on silica gel and loaded onto a column for further separation. Fractions **F_6_**–**F_9_** were subjected to repeated silica gel columns and eluted with EtOAc: MeOH (95:5) isocratically to afford compound **5** as white crystals (36.0 mg). Fractions **F_12_**–**F_14_** were chromatographed on silica gel using 100% EtOAc, isocratically. This resulted in eleven subfractions, from which the combined subfractions five–seven were further purified on a silica gel column eluting isocratically with DCM-MeOH (95:5) to give compound **7** (45.8 mg). Fractions **F_16_**, **F_18_**–**F_20_**, and **F_22_** were each purified in Sephadex LH-20 using 95% ethanol as the eluent, isocratically. Fraction **F_16_** yielded compound **2** (20.5 mg), **F_18_**–**F_20_** afforded compound **8** (202.6 mg), and compound **3** (18.0 mg) was obtained from **F_22_**. Fraction **F_24_** was chromatographed on a silica gel column using the DCM: MeOH (90:10, 85:15) gradient and a total of eight subfractions were obtained, from which four–six were further purified on silica gel to give compound **4** (12.2 mg).

### 3.4. Antityrosinase Inhibition Assay

The skin enzymatic inhibitory assay was executed during the study following the approach used by Curto et al. [39] and Nerya et al. [40]. Extracts and isolates were dissolved in dimethyl sulfoxide (DMSO) to a final concentration of 20 mg/mL. The stock solutions were then diluted to 100 and 200 µg/mL in 50 mM of potassium phosphate buffer (pH 6.5). ‘Kojic acid’ was used as a control drug [41]. In the wells of a 96-well plate, 70 µL of each sample dilution was combined with 30 µL of tyrosinase (500 units/mL in phosphate buffer) in triplicate. After incubation at room temperature for 5 min, 110 µL of the substrate (2 mM L-tyrosine) was added to each well. Final concentrations of the extracts and isolated samples and positive controls ranged from 0.2 to 1000 µg/mL. Incubation commenced for 30 min at room temperature by measuring the absorbance at 490 nm with the AccuReader M965 Metertech (V1.11). Equation 1 was employed in determining the percentage of tyrosinase inhibition.

Equation (1): percentage of tyrosinase inhibition.
(1)$$\%\;\mathrm{of}\;\mathrm{tyrosinase}\;\mathrm{inhibition}=\frac{\left(A_{control}\;-\;A_{\;blank\;1}\right)\;-\;\left(A_{sample}\;-\;A_{\;blank\;2}\right)}{\left(A_{\;control}\;-\;A_{\;blank}\right)}\times 100$$
where ***A_control_*** is the absorbance of the control with the enzyme, ***A*_*blank* 1_** is the absorbance of the control without the enzyme, ***A_sample_*** is the absorbance of the test sample with the enzyme, and ***A*_*blank* 2_** is the absorbance of the test sample without the enzyme.

## 4. Conclusions

The phytochemical investigation of *Protea cynaroides* afforded eight compounds, from which compounds **1**, **2**, **3**, and **4** were reported for the first time from a natural source, while **8** was isolated for the first time from *Protea cynaroides*. Compounds **7** and **8** exhibited potent competitive tyrosinase inhibition against the L-tyrosine substrate, while **6** and **5** demonstrated weak activity. Good anti-tyrosinase activity exhibited by two of these compounds suggests the potential exploration of *P. cynaroides* in the cosmetic and pharmaceutical industries upon further biological and clinical investigations. This study highlights the importance of *P*. *cynaroides*, the national plant of South Africa, as a medicinal plant with therapeutic potential. This is the first scientific report on the bio-evaluation of tyrosinase inhibitory activities of *Protea cynaroides.*

## Figures and Tables

**Figure 1 plants-11-01751-f001:**
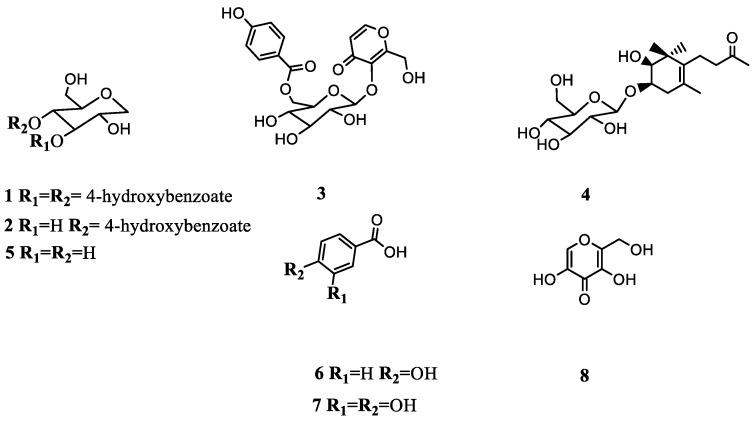
Chemical structures of the isolated compounds (**1**–**8**) from *P. cynaroides*.

**Figure 2 plants-11-01751-f002:**
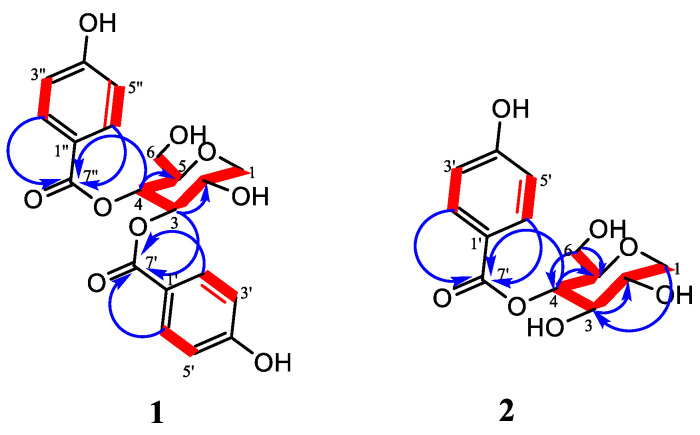
Key HMBC (→) and COSY (▬) correlations of compounds **1** and **2**.

**Figure 3 plants-11-01751-f003:**
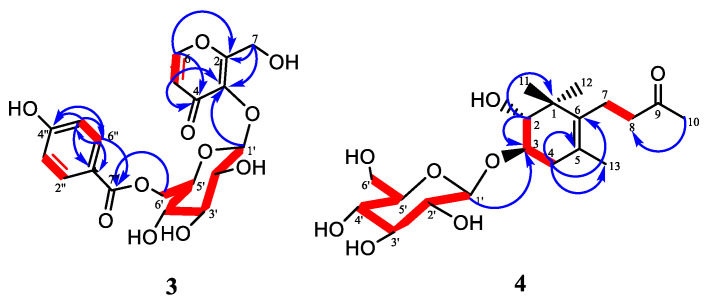
Key HMBC (→) and COSY (▬) correlations of compounds **3** and **4**.

**Figure 4 plants-11-01751-f004:**
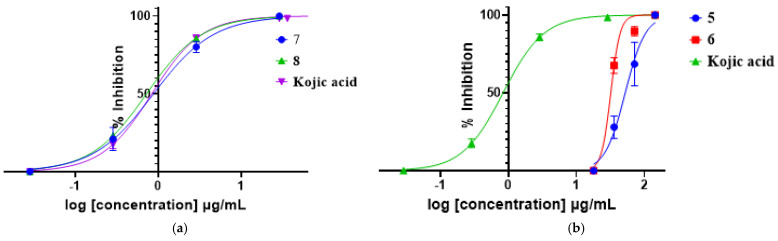
Inhibitory effects on tyrosinase activity of compounds **7** and **8** (**a**), **5** and **6** (**b**), and the positive control, kojic acid.

**Table 1 plants-11-01751-t001:** ^1^H and ^13^C NMR data of compounds **1** and **2**.

	1 in DMSO-*d*_6_		2 in CD_3_OD	
**Position**	**ẟ_H_ (*mult*, *J*)**	**ẟ_C_**	**ẟ_H_ (*mult*, *J*)**	**ẟ_C_**
**1a**	3.33 (*dd*, 11.1, 5.5)	69.9 *t*	3.16 (*t*, 10.6)	71.1 *t*
**1b**	3.94 (*dd*, 11.1, 5.5)		3.89 (*dd*, 11.2, 4.6)	
**2**	3.76 (*m*)	68.3 *d*	3.51 (*m*) *	71.7 *d*
**3**	5.27 (*t*, 9.3)	77.5 *d*	3.51 (*m*) *	77.9 *d*
**4**	5.05 (*t*, 9.7)	69.7 *d*	4.81 (*t*, 9.3)	73.0 *d*
**5**	3.59 (*ddd*, 4.2, 2.3, 2.2)	79.5 *d*	3.38 (*m*) *	81.0 *d*
**6a**	3.38 (*dd*, *)	61.2 *t*	3.40 (*m*) *	62.9 *t*
**6b**	3.45 (*dd*, *)		3.49 (*m*)	
**1′**	-	120.8 *s*	-	122.1 *s*
**2′**	7.71 (*d*, 8.8)	132.0 *d*	7.83 (*d*, 8.8)	133.2 *d*
**3′**	6.78 (*d*, 8.8)	115.6 *d*	6.75 (*d*, 8.8)	116.3 *d*
**4′**	-	162.3 *s*	-	163.8 *s*
**5′**	6.78 (*d*, 8.8)	115.6 *d*	6.75 (*d*, 8.8)	116.3 *d*
**6′**	7.71 (*d*, 8.8)	132.0 *d*	7.83 (*d*, 8.8)	133.2 *d*
**7′**	-	165.6 *s*		167.7 *s*
**1″**	-	120.1 *s*		
**2″**	7.68 (*d*, 8.8)	131.9 *d*		
**3″**	6.78 (*d*, 8.8)	115.7 *d*		
**4″**	-	162.6 *s*		
**5″**	6.78 (*d*, 8.8)	115.7 *d*		
**6″**	7.68 (*d*, 8.8)	131.9 *d*		
**7″**	-	165.0 *s*		

* Peaks overlapped, assignments based on DEPT, COSY, HSQC, and HMBC experiments.

**Table 2 plants-11-01751-t002:** ^1^H and ^13^C NMR data of compounds **3** and **4**.

	3 in CD_3_OD			4 in DMSO-*d*_6_	
**Position**	**ẟ_H_ (*mult*, *J*)**	**ẟ_C_**	**Position**	**ẟ_H_ (*mult*, *J*)**	**ẟ_C_**
**2**	-	163.9 *s*	**1**	-	42.2 *s*
**3**	-	142.3 *s*	**2**	3.16 (*m*)	76.9 *d*
**4**	-	177.4 *s*	**3**	3.70 (*d*, 6.7) *	75.7 *d*
**5**	6.31 (*d*, 5.6)	117.7 *d*	**4a**	2.00 (*dd*, 17.2, 7.1) *	37.9 *t*
**6**	7.97 (*d*, 5.6)	157.5 *d*	**4b**	2.31 (*dd*, 17.4, 6.8)	
**7a**	4.69 (*d*, 14.1)	57.7 *t*	**5**	-	123.9 *s*
**7b**	4.47 (*d*, 14.1)		**6**	-	136.0 *s*
**1′**	4.94 (*d*, 7.7)	104.0 *d*	**7a**	2.09 (*m*)	22.2 *t*
**2′**	3.41 (*m*)	75.4 *d*	**7b**	2.16 (*m*)	
**3′**	3.45 (*m*)	77.8 *d*	**8**	2.45 (*t*, 8.1)	43.9 (CH_2_)
**4′**	3.43 (*m*)	71.7 *d*	**9**	-	208.9 *s*
**5′**	3.56 (*m*)	76.2 *d*	**10**	2.07 (*s*)	30.1 (CH_3_)
**6′a**	4.58 (*dd*, 11.8, 2.2)	64.4 *s t*	**11**	1.01 (*s*)	25.6 (CH_3_)
**6′b**	4.45 (*dd*, 11.8, 2.2)		**12**	0.86 (*s*)	22.0 (CH_3_)
**1″**	-	122.1 *s*	**13**	1.56 (*s*)	19.5 (CH_3_)
**2″**	7.84 (*d*, 8.8)	130.0 *d*	**1′**	4.28 (*d*, 7.8)	101.4 *d*
**3″**	6.83 (*d*, 8.8)	116.4 *d*	**2′**	3.00 (*m*)	73.7 *d*
**4″**	-	163.7 *s*	**3′**	3.15 (*m*)	77.2 *d*
**5″**	6.83 (*d*, 8.8)	116.4 *d*	**4′**	3.05 (*m*)	70.4 *d*
**6″**	7.84 (*d*, 8.8)	130.0 *d*	**5′**	3.16 (*m*)	76.8 *d*
**7″**	-	167.9 *s*	**6′a**	3.44 (*m*)	61.4 *t*
			**6′b**	3.65 (*m*)	

* Peaks overlapped, assignments based on DEPT, COSY, HSQC, and HMBC experiments.

**Table 3 plants-11-01751-t003:** Tyrosinase inhibitory activity of the isolated compounds.

Extracts/ Compounds	IC_50_(µg/mL) ± SD
TE *Bu*-F **1**	85.2 75.5 NA *
**2**	NA *
**3**	NA *
**4**	NA *
**5**	274.5 ± 2.12
**6**	149.2 ± 1.06
**7**	0.8776 ± 0.12
**8**	0.7215 ± 0.09
Kojic acid	0.8347 ± 0.093

* NA: not active at the tested concentrations; TE: total extract; *Bu*-F: butanol fraction.

## Data Availability

The raw data presented in this study are available on request from the corresponding author(s).

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
