# Peer review of "Characterization of Four New Compounds from *Protea cynaroides* Leaves and Their Tyrosinase Inhibitory Potential"

_plants, 2022, doi:10.3390/plants11131751_

Round 1
Reviewer 1 Report
It is an interesting manuscript that reports the isolation and characterization of eight compounds from Protea cynaroides leaves (including four new) and investigation of their tyrosinase-inhibiting activity. There are some remarks that should be considered and addressed before the publication of the manuscript.
1. The title of the manuscript should be revised.
2. The sentence in the Abstract “This study is the first chemical investigation of P. cynaroides and” is misleading because there are other study on the phytochemical composition of the plant. Revise it.
3. The study lacks clear and well defined aim.
4. Section 3.4 Bio evaluation should be removed and replaced with just “Antityrosinase inhibition assay”.
5. In general, the discussion of the obtained results in regards to previous phytochemical studies is not sufficient.
6. Introduction shows that there many different phenolics have been identified in the investigated specie. However, they were not isolated in the current study. This fact deserve an in-depth discussion.
7. The English of the manuscript should be checked and revised.
Reviewer 2 Report
Plants belonging to the Proteaceae family were screened for tyrosinase activity. Extracts from Protea cynaroides
showed activity. 8 Substances could be isolated from extracts. Structure of 4 substances could be determined by 1D (1H, 13C, and DEPT 135) and 2D-NMR experiments (HSQC and HMBC), FTIR and LC-MS data. The other 4 substances were identified by comparison of spectroscopic data to literature data.
2 substances show reasonably mushroom tyrosinase inhibition effects.
The manuscript clearly written.
However the inhibition curves of Kojic acid show different courses in Figure 4 and the resulting IC50 values should differ for 2 orders of magnitude. Using three data points the curve of the Kojic acid fit in Figure 4b is wishful thinking.
Author Response
Response to Reviewer’s Comments
Dear Editor,
Coauthors and I very much appreciated the encouraging, critical and constructive comments on this manuscript by the reviewer. The comments have been very thorough and useful in improving the manuscript. We strongly believe that the comments and suggestions have increased the scientific value of revised manuscript by many folds. We have taken them fully into account in revision. We are submitting the corrected manuscript with the suggestion incorporated the manuscript. The manuscript has been revised as per the comments given by the reviewer, and our responses to all the comments are as follows:
Reviewer #2
Original comments of the reviewer:
- The inhibition curves of Kojic acid show different courses in Figure 4 and the resulting IC50 values should differ for 2 orders of magnitude. Using three data points the curve of the Kojic acid fit in Figure 4b is wishful thinking
Reply by the author(s):
Thank you for the comment. We have revised the inhibition curves as shown in the figure 4b

Reviewer 3 Report
In the present manuscript, the authors isolated eight compounds from Protea cynaroides. They also determine their tyrosinase inhibitory activities. Such research is of great importance in the use of substances of natural origin that exhibit some biological activity. The isolation and characterization of substances from natural extracts allows the individual use of the specific substance in practice. This leads to the possibility of lowering the therapeutic doses of the substances used and reducing the toxic effect and side effects, because substances from the extract that were contained in it but do not have the desired biological effect are eliminated.
Therefore, I believe that the study is extremely important for the characterization and application of the components of Protea cynaroides in future studies.
My remarks to the authors are just some technical omissions.
1. The abstract is not designed according to the requirements of the editors:
The abstract should be a single paragraph and should follow the style of structured abstracts, but without headings: 1) Background; 2) Methods; 3) Results: and 4) Conclusion.
2. Some sentences at the end of paragraphs appear incomplete due to an lack of a full stop (lines 46, 240, 243, 257 and 364).
3. After the title of Figure 1 there are two full stops (line 96). In the same title P. cynaroids has a spelling mistake, it must be P. cynaroides and italicized.
4. The paragraph "Author Contributions" is not filled in at the end of the manuscript. The participation of each of the authors should be indicated - how he/she contributed to the existence of the manuscript (see instructions for authors).
5. At the end of the manuscript, the "Data Availability Statement" contains literally the requirements of the editors to the authors, without real information. My suggestion is to drop it if there is no information; if there is information, indicate where data supporting the reported results can be found; or indicate that is "Not applicable".
Author Response
Response to Reviewer’s Comments
Dear Editor,
Coauthors and I very much appreciated the encouraging, critical and constructive comments on this manuscript by the reviewer. The comments have been very thorough and useful in improving the manuscript. We strongly believe that the comments and suggestions have increased the scientific value of revised manuscript by many folds. We have taken them fully into account in revision. We are submitting the corrected manuscript with the suggestion incorporated the manuscript. The manuscript has been revised as per the comments given by the reviewer, and our responses to all the comments are as follows:
Reviewer #3
Original comments of the reviewer:
- The abstract is not designed according to the requirements of the editors:
The abstract should be a single paragraph and should follow the style of structured abstracts, but without headings: 1) Background; 2) Methods; 3) Results: and 4) Conclusion.
Reply by the author(s):
We are thankful to the reviewer for the comments. The abstract has been modified accordingly.
Original comments of the reviewer:
- Some sentences at the end of paragraphs appear incomplete due to a lack of a full stop (lines 46, 240, 243, 257 and 364).
Reply by the author(s):
Thank you to the reviewer. We have carefully read the article and made the corrections.
Original comments of the reviewer:
- After the title of Figure 1 there are two full stops (line 96). In the same title P. cynaroides has a spelling mistake, it must be P. cynaroides and italicized.
Reply by the author(s):
We have noted the comments. The errors have rectified.
Original comments of the reviewer:
- The paragraph "Author Contributions" is not filled in at the end of the manuscript. The participation of each of the authors should be indicated - how he/she contributed to the existence of the manuscript (see instructions for authors).
Reply by the author(s):
We acknowledge the reviewers’ comments and we have applied the suggestions.
Original comments of the reviewer:
- At the end of the manuscript, the "Data Availability Statement" contains literally the requirements of the editors to the authors, without real information. My suggestion is to drop it if there is no information; if there is information, indicate where data supporting the reported results can be found; or indicate that is "Not applicable".
Reply by the author(s):
Thank you for the comment. We have attended to this in the manuscript.

Reviewer 4 Report
The MS entitled ``Characterization of Four New Compounds and Tyrosinase Inhibitors Isolated from Protea cynaroides Leaves`` cannot be published in the current form, the below issues should be addressed
-I suggest modifying the title to ``Characterization of Four New Compounds from Protea cynaroides Leaves and Their Tyrosinase Inhibitory Potential``.
-English editing is needed.
-The plant family name should be added in the abstract and keywords. Remove, kojic acid derivatives from keywords
- The plant part and extract type should be added in the abstract.
- The type of assay, control name, and control result should be added in the abstract.
- in line 30-31, a refence should be added.
- in line 36-39, a refence should be added.
- in line 47, a refence should be added.
-P. cynaroides should be italic through the whole MS.
-Remove the compounds` quantities from discussion.
-ESI-MS should be corrected to HRESI-MS.
-in line 105, 4-hyroxybenzoate should be corrected to trisubstituted benzene moiety.
-The H-H COSY and HMBC correlations that established each structure subunits should be discussed in detail.
-Compound 8 is a known compound, no need for discussing its structural elucidation.
-A voucher specimen number should be included.
- Conclusion should be modified, no need to mention the IC50 again in conclusion.
- Figure 1 needs improvement the atoms stick with each other and the writing under each structure should unify the font size and type.
Author Response
Response to Reviewer’s Comments
Dear Editor,
Coauthors and I very much appreciated the encouraging, critical and constructive comments on this manuscript by the reviewer. The comments have been very thorough and useful in improving the manuscript. We strongly believe that the comments and suggestions have increased the scientific value of revised manuscript by many folds. We have taken them fully into account in revision. We are submitting the corrected manuscript with the suggestion incorporated the manuscript. The manuscript has been revised as per the comments given by the reviewer, and our responses to all the comments are as follows:
Examiner #4
Original comments of the reviewer:
- I suggest modifying the title to ``Characterization of Four New Compounds from Protea cynaroides Leaves and Their Tyrosinase Inhibitory Potential``
Reply by the author(s):
We thank the reviewer for the suggestion. The title has been modified.
Original comments of the reviewer:
- English editing is needed.
Reply by the author(s):
We acknowledge the reviewers’ comments. We have carefully read the article and corrected the grammatical and linguistic errors as suggested.
Original comments of the reviewer:
- The plant family name should be added in the abstract and keywords. Remove, kojic acid derivatives from keywords.
Reply by the author(s):
We appreciate the reviewer’s suggestions. The changes have been applied.
Original comments of the reviewer:
- The plant part and extract type should be added in the abstract.
Reply by the author(s):
Thank you for this comment. We have revised the abstract as recommended.
Original comments of the reviewer:
- The type of assay, control name, and control result should be added in the abstract.
Reply by the author(s):
We appreciate the suggestion by the reviewer. The abstract has been modified.
Original comments of the reviewer:
- In line 30-31, a refence should be added.
Reply by the author(s):
Thank you for the comment. The reference has been added.
Original comments of the reviewer:
- In line 36-39, a refence should be added.
Reply by the author(s):
We are thankful for this comment. The reference has been added.
Original comments of the reviewer:
- In line 47, a refence should be added.
Reply by the author(s):
We acknowledge this comment. The reference has been added.
Original comments of the reviewer:
- P. cynaroides should be italic through the whole MS.
Reply by the author(s):
Thank you for the comment. We have carefully read the article and we have corrected the mistakes.
Original comments of the reviewer:
- Remove the compounds` quantities from discussion.
Reply by the author(s):
Thank you for the comment. The changes were applied.
Original comments of the reviewer:
- ESI-MS should be corrected to HRESI-MS.
Reply by the author(s):
Thank you for the comment. This has been corrected.
Original comments of the reviewer:
- In line 105, 4-hyroxybenzoate should be corrected to trisubstituted benzene moiety
Reply by the author(s):
Thank you for the comment. This has been corrected.
Original comments of the reviewer:
- The H-H COSY and HMBC correlations that established each structure subunits should be discussed in detail.
Reply by the author(s):
We have noted the reviewer’s comments and have revised the discussion.
Original comments of the reviewer:
- Compound 8 is a known compound, no need for discussing its structural elucidation.
Reply by the author(s):
Thank you for the comment. We have removed the discussion of compound 8.
Original comments of the reviewer:
- A voucher specimen number should be included.
Reply by the author(s):
We acknowledge this comment by the reviewer. The voucher specimen number has been included.
Original comments of the reviewer:
- Conclusion should be modified, no need to mention the IC50 again in conclusion.
Reply by the author(s):
Thank you for the comment. The conclusion has been modified.
Original comments of the reviewer:
- Figure 1 needs improvement the atoms stick with each other and the writing under each structure should unify the font size and type.
Reply by the author(s):
Thank you for the comment. Figure 1 has been improved.

Round 2
Reviewer 1 Report
Corrections made by authors are satisfactory. As a result the manuscript was significantly improved and could be accepted for publication.
Author Response
Thank you so much.
Reviewer 2 Report
Dear authors,
I have rejected the last version because of a mistake in Fig.4. The present version still contains this error.
If you follow the curve of Kojic acid in Fig. 4 (a) 50 % Inhibition is reached at "0" on the concentration axis. If you look at Fig. 4 (b) 50 % Inhibition is reached at "1" on the concentration axis, i.e. a tenfold higher concentration of Kojic acid. There is something wrong in the presentation of your data!
Reviewer 4 Report
No comments
Author Response
Thank you so much.